Differences in meiofauna communities with sediment depth are greater than habitat effects on the New Zealand continental margin: implications for vulnerability to anthropogenic disturbance

Rosli Norliana 1 2 3 rosli.norliana@gmail.com
Leduc Daniel 2
Rowden Ashley A. 2
Clark Malcolm R. 2
Probert P. Keith 1
Berkenbusch Katrin 1 4
Neira Carlos 5
1 Department of Marine Science, University of Otago , Dunedin , New Zealand
2 National Institute of Water and Atmospheric Research (NIWA) , Wellington , New Zealand
3 Department of Biology, Faculty Science & Mathematics, Universiti Pendidikan Sultan Idris , Tg. Malim, Perak , Malaysia
4 Dragonfly Data Science , Wellington , New Zealand
5 Integrative Oceanography Division, Scripps Institution of Oceanography , La Jolla, California , United States
Reimer James
Electronic publication date: 2016 Jul 5
Publication date: 2016
Volume: 4
Electronic Location ID: e2154
Received 2016 Mar 27; Accepted 2016 May 30
Copyright: © 2016 Rosli et al.
Copyright year: 2016
Copyright holder: Rosli et al.
License: This is an open access article distributed under the terms of the Creative Commons Attribution License, which permits unrestricted use, distribution, reproduction and adaptation in any medium and for any purpose provided that it is properly attributed. For attribution, the original author(s), title, publication source (PeerJ) and either DOI or URL of the article must be cited.
License URL: https://creativecommons.org/licenses/by/4.0/

Keywords: Meiofauna, Canyon, Seamount, Seep, Slope, Fishing, New Zealand

Funding: New Zealand Ministry for Business, Innovation and Employment CO1X0906 NOAA Grants NA17RJ1231 and NA05417076 University of Otago (New Zealand) Universiti Pendidikan Sultan Idris (Malaysia) Kementerian Pendidikan Tinggi (Malaysia) This study was part of NIWA’s research project ‘Impact of resource use on vulnerable deep-sea communities’ funded by the New Zealand Ministry for Business, Innovation and Employment (CO1X0906). Voyage TAN0616 (RENEWZ) was funded by NOAA grants (NA17RJ1231 and NA05417076) and NIWA. N. Rosli received PhD research funding from the University of Otago (New Zealand), Universiti Pendidikan Sultan Idris (Malaysia), and Kementerian Pendidikan Tinggi (Malaysia). The funders had no role in study design, data collection and analysis, decision to publish, or preparation of the manuscript.

==============================
Studies of deep-sea benthic communities have largely focused on particular (macro) habitats in isolation, with few studies considering multiple habitats simultaneously in a comparable manner. Compared to mega-epifauna and macrofauna, much less is known about habitat-related variation in meiofaunal community attributes (abundance, diversity and community structure). Here, we investigated meiofaunal community attributes in slope, canyon, seamount, and seep habitats in two regions on the continental slope of New Zealand (Hikurangi Margin and Bay of Plenty) at four water depths (700, 1,000, 1,200 and 1,500 m). We found that patterns were not the same for each community attribute. Significant differences in abundance were consistent across regions, habitats, water and sediment depths, while diversity and community structure only differed between sediment depths. Abundance was higher in canyon and seep habitats compared with other habitats, while between sediment layer, abundance and diversity were higher at the sediment surface. Our findings suggest that meiofaunal community attributes are affected by environmental factors that operate on micro- (cm) to meso- (0.1–10 km), and regional scales (> 100 km). We also found a weak, but significant, correlation between trawling intensity and surface sediment diversity. Overall, our results indicate that variability in meiofaunal communities was greater at small scale than at habitat or regional scale. These findings provide new insights into the factors controlling meiofauna in these deep-sea habitats and their potential vulnerability to anthropogenic activities.

Introduction

Continental margins comprise a variety of topographically-defined habitats such as canyons, seamounts and slopes, as well as chemically-defined habitats such as cold seeps and hydrothermal vents (Levin et al., 2010). Canyons are complex topographic features that influence local hydrodynamic regimes, and thus sediment transport and accumulation (García et al., 2008). The resulting changes in physico-chemical characteristics and organic enrichment in the sediments have been linked to high variation in infaunal benthic community structure (Baguley et al., 2006; de Stigter et al., 2007; García et al., 2008; Romano et al., 2013). Seamounts, which are defined as elevated features that include knolls, pinnacles and hills where the elevation can be as low as 100 m (Clark et al., 2010; Pitcher et al., 2007), can affect surrounding flow conditions resulting in enhanced currents, eddies, up- and down-welling and closed retention cells (Bashmachnikov, Loureiro & Martins, 2013; White et al., 2007). These modified flow conditions increase vertical mixing, spatial variation in sedimentation processes, and the distribution of food resources (Bongiorni et al., 2013; Levin & Dibacco, 1995; Zeppilli et al., 2013). These and other factors can result in distinct benthic communities on seamounts (Bongiorni et al., 2013; Zeppilli et al., 2014). Cold seeps are characterised by the flow of reduced chemical compounds (e.g., methane, sulphur) from the subsurface to the seafloor (Lampadariou et al., 2013; Levin, 2005; Van Gaever et al., 2009). The emission of reduced fluids results in a broad range of geological and sedimentary structures (e.g., gas seepage, microbial mat, pockmarks) (Judd, Jukes & Leddra, 2002; Levin, 2005), which increase small-scale variability in the sediment, thus providing a variety of habitats for infauna that differ from ‘background’ habitats (Levin & Mendoza, 2007). Hydrothermal vents are localized areas of the seabed where heated and chemically modified seawater exits the seafloor as diffuse or focused flow (Van Dover, 2014). Vent ecosystems are typically dominated by benthic invertebrate taxa that host symbiotic, chemoautotrophic microorganisms, and the infauna of hydrothermally ‘active’ sediments has been shown to differ from that of ‘inactive’ sediments (Levin et al., 2009).

Meiofauna are the most abundant infauna in deep-sea sediments, with nematodes being the most abundant taxon (Heip, Vincx & Vranken, 1985; Vanreusel et al., 2010). Studies of meiofaunal communities in the deep sea have focused on canyon and adjacent slope habitats (Bianchelli et al., 2008; Danovaro et al., 2009; Soetaert & Heip, 1995; Soltwedel et al., 2005), and few comparative studies have included seamount (Zeppilli et al., 2013) or cold seep habitats (Pape et al., 2011; Robinson et al., 2004). Vanreusel et al. (2010) provided the first comprehensive comparison of nematode communities among multiple deep-sea habitats (e.g., canyon, seamounts, seep and vent), and showed that different habitats harbour distinct nematode communities and therefore contribute to overall deep-sea nematode diversity.

Although our understanding of meiofaunal community structure of deep-sea habitats is growing, there is remaining uncertainty as meiofauna are not considered in a number of biodiversity studies and are generally poorly studied (particularly in the deep sea) compared to larger macrofauna (Zeppilli et al., 2015). In addition, a more rigorous test of habitat effects on meiofaunal communities requires comparisons that avoid the potential influence of geographical distance on community patterns. Knowledge of meiofaunal distribution and connectivity between different habitats is essential for understanding ecological processes, and for assessing the vulnerability of benthic communities to anthropogenic disturbance. There have been concerns about the potential threats of anthropogenic activities on the diversity and function of deep-sea ecosystems (Pusceddu et al., 2014; Ramirez-Llodra et al., 2011; Van Dover, 2014), as technological advances make these habitats more accessible (Benn et al., 2010; Levin & Sibuet, 2012). For example, industrial fisheries are expanding and moving into deeper waters (Pitcher et al., 2010; Pusceddu et al., 2014), and seabed mining in the deep sea is expected to begin in the near future (Hein et al., 2013; Ramirez-Llodra et al., 2015).

Physical disruption of habitat by bottom trawling can have pronounced effects on deep-sea soft sediment communities (Pusceddu et al., 2014). Trawling generally has a negative impact on macro-infaunal communities (Hansson et al., 2000; Hinz, Prieto & Kaiser, 2009), whereas studies of meiofauna reveal inconsistent results. To date, studies from shallow water habitats suggest that trawling may have a positive (Liu et al., 2011; Pranovi et al., 2000), negative (Hinz et al., 2008; Schratzberger & Jennings, 2002), or only minor impact (Lampadariou, Hatziyanni & Tselepides, 2005; Liu, Cheung & Shin, 2009; Schratzberger, Dinmore & Jennings, 2002) on meiofaunal communities. The only meiofaunal study conducted in the deep sea (Pusceddu et al., 2014) showed a negative effect of trawling on meiofaunal communities. Meiofauna, and nematodes in particular, are generally considered to be more resilient to physical disturbance than larger organisms because they are less likely to be killed and can recover more quickly (Leduc & Pilditch, 2013; Schratzberger, Dinmore & Jennings, 2002; Whomersley et al., 2009). Nevertheless, bottom trawling can also have indirect impacts on sediment communities through the modification of sediment physical characteristics and distribution of organic matter, which can lead to potentially long-term changes in benthic communities (Martín et al., 2014; Pusceddu et al., 2014).

Because of their smaller size, meiofauna tend to respond to micro-scale (cm) variability of environmental conditions in surface and subsurface sediment layers (Ingels et al., 2011; Ingels, Tchesunov & Vanreusel, 2011; Soetaert et al., 1997). Ingels & Vanreusel (2013) showed that most of the variability in nematode community structure occurs at micro (cm) rather than larger spatial scales (10–100 km). Decline in meiofaunal densities with sediment depth is probably the most pervasive gradient observed in marine sediments (Ingels et al., 2009; Soltwedel et al., 2005; Van Gaever et al., 2006; Vanaverbeke et al., 1997), with the vertical distribution of meiofauna in the sediments mainly controlled by decreasing food and oxygen availability in subsurface sediments (Giere, 2009; Moens et al., 2014; Vanaverbeke et al., 1997; Vanreusel et al., 1995). Meiofaunal diversity is typically highest in surface sediment and decreases in deeper sediments where nematodes become dominant (Danovaro, Gambi & Della Croce, 2002; Schmidt & Martínez Arbizu, 2015). The more abundant and diverse meiofaunal communities of surface sediments are more exposed to disturbance than subsurface communities, and may therefore be affected more by physical disturbance. Studies aiming to uncover the processes driving the composition of deep-sea meiofaunal communities, including potential physical disturbance, should therefore include examination of variation at these smaller scales.

The main objectives of this study were to: (1) compare meiofaunal community attributes (abundance, diversity and community structure) in surface (0–1 cm) and subsurface (1–5 cm) sediment layers among deep-sea habitats; (2) describe relationships between environmental variables (i.e., water depth, sediment characteristics, topography, food availability), bottom trawling and community attributes of meiofaunal communities; (3) assess the relative vulnerability of meiofaunal communities among habitats, and between surface and subsurface sediment layers.

Material and Methods

Study area and sampling design

The study area comprised two regions: Hikurangi Margin and Bay of Plenty of New Zealand (Fig. 1). These two regions were selected because each encompasses a range of benthic habitats within a restricted geographic area, thus facilitating comparisons between associated faunas that were not confounded by distance. The Hikurangi Margin study region is located to the north-east of the South Island, hosts many submarine canyons on its continental slope, and also includes other deep-sea habitats such as seamounts, and cold seeps (Mountjoy, Barnes & Pettinga, 2009; Ruff et al., 2013). The Bay of Plenty study region, located to the north-east of North Island, also includes slope, canyon and seamount habitats, with hydrothermal vents on some seamounts (Wysoczanski & Clark, 2012). The Hikurangi Margin hosts significant fisheries, including hoki (Macruronus novaezelandiae), alfonsino (Beryx splendens) and orange roughy (Hoplostethus altanticus) which occur across all habitats (Clark, 1995). This area is also of potential interest for drilling gas hydrate deposits (Pecher & Henrys, 2003). The Bay of Plenty region is subject to some deep-sea trawl fisheries, including orange roughy, black cardinal fish (Epigonus telescopus) and alfonsino (Beryx decadactylus) (Clark & O’Driscoll, 2003), and is of potential interest for mining of seafloor massive sulphide deposits (Boschen et al., 2013).

Figure 1 Map showing sampling sites and stations in the Bay of Plenty (BoP, (A)) and Hikurangi Margin (HIK, (B)) study regions and their relative locations in New Zealand (inset).

Scale bar applies to both regional maps. Not all sites and stations could be sampled in the present study. Refer to Table 1 for a list of sites and stations where meiofaunal samples were obtained. The blue strips in the top panel show multibeam lines where bathymetry is more detailed than the underlying pale blue.

Sampling sites have been previously described by Bowden et al. (2016). Sampling was conducted at slope, canyon and seamount sites from RV Tangaroa during National Institute of Water and Atmospheric Research (NIWA) voyage TAN1004 (April 2010) on the Hikurangi Margin, and voyage TAN1206 (April 2012) in the Bay of Plenty (Fig. 1). The samples were collected under Special Permit (542) issued by the Ministry for Primary Industries pursuant to section 97(1) of the Fisheries Act 1996. Fishing intensity was included as a variable in the analysis (see below) to account for the possible influence of anthropogenic disturbance on the main analysis. Trawl effort data for the period July 1980 to March 2011 were sourced from the trawl database of the New Zealand Ministry for Primary Industries. Sampling was undertaken at four water depth strata (700, 1,000, 1,200 and 1,500 m) at each habitat site to incorporate the effects of water depth in the statistical analyses and provide a more robust evaluation of any habitat effect on community structure. At Hikurangi Margin, meiofauna could not be sampled at some sites/depths, whereas in Bay of Plenty, the limited occurrence of soft sediment prevented the sampling on seamount and vent habitats. The limited data from these sites were not included in the analysis (Table 1). At each sampling station, a towed video camera frame was deployed along transects to ascertain the type of substratum and benthic megafauna before the water column and seafloor was disturbed by sampling gear. Deployment of the multicorer, which targeted soft sediment substrates, was directed based on information from multibeam echo-sounder (MBES) bathymetric maps and observations from the video transects.

Table 1 List of sampling sites for Hikurangi Margin (TAN1004) and Bay of Plenty (TAN1206) (see Bowden et al., 2016).

Full names for named features are: Campbell Canyon, Honeycomb Canyon, Pahaua Canyon, Tauranga Canyon, White Island Canyon and Runaway Sea Valley. Unnamed seamount features are labelled according to the registration number of NIWA New Zealand seamounts database (e.g., ‘SMT_310’).

Region	Voyage	Sampling date	Habitat	Site	Strata	Station	Depth (m)	Latitude (S)	Longitude (E)	N	
Hikurangi margin	TAN1004	April 2010	Slope	1	700	124	690	41.9857	174.6982	2	
1,500	128	1,420	42.0485	174.7000	1	
2	1,000	4	1,046	41.6837	175.6642	3	
1,200	76	1,282	41.6833	175.6500	2	
1,500	10	1,561	41.7170	175.6748	2	
3	700	44	728	41.5258	175.8003	3	
1,000	41	942	41.5475	175.8398	3	
1,200	38	1,121	41.5937	175.8532	3	
1,500	17	1,514	41.6288	175.8682	2	
1,500	19	1,553	41.6270	175.8637	1	
Canyon	Pahaua	700	31	730	41.4962	175.6828	3	
1,000	27	1,013	41.4983	175.7043	3	
1,200	22	1,188	41.5100	175.7187	3	
1,500	12	1,350	41.5508	175.7250	3	
Honeycomb	700	58	670	41.4080	175.8977	3	
1,000	53	948	41.4563	175.8970	1	
1,200	62	1,171	41.4760	175.9477	3	
Campbell	700	92	683	41.8922	174.6347	2	
1,000	97	1,011	41.9458	174.6173	1	
1,000	98	1,012	41.9277	174.6165	2	
1,200	127	1,177	42.1228	174.5397	1	
1,500	126	1,495	42.1422	174.5492	3	
Seamount	310	700	69	670	41.3353	176.1882	3	
1,000	72	985	41.3657	176.1958	3	
766	1,000	130	894	42.1363	174.5737	1	
1,500	129	1,456	42.1345	174.5860	1	
1,500	132	1,453	42.1345	174.5850	1	
South tower	TAN0616	Nov 2006	Seep	Opouawe bank	1,000	84	1,053	41.7832	175.4007	2	
1,000	86	1,050	41.782	175.402	2	
1,000	116	1,049	41.7885	175.4075	2	
1,000	118	1,051	41.7893	175.4072	2	
North tower					1,000	112	1,054	41.0782	175.4013	2	
1,000	123	1,051	41.079	175.4075	2	
Bay of plenty	TAN1206	April 2012	Slope	1	700	2	699	37°10.14	176°39.58	3	
1,000	5	998	37°06.74	176°43.86	3	
1,200	9	1,193	37°03.48	176°48.38	3	
1,500	13	1,501	37°55.35	176°58.74	3	
2	700	52	710	37°30.26	177°37.19	3	
1,000	49	1,004	37°25.90	177°37.55	3	
1,200	44	1,202	37°21.95	177°37.57	3	
1,500	42	1,501	37°14.50	177°37.86	3	
3	700	185	726	37°22.84	178°01.92	3	
1,000	181	998	37°20.56	178°01.71	3	
1,200	178	1,196	37°19.01	178°01.42	3	
1,500	175	1,494	37°15.66	178°00.23	3	
Canyon	Tauranga	700	125	697	37°28.48	176°45.51	3	
1,000	118	1,083	37°20.00	176°57.72	3	
1,200	111	1,221	37°15.05	176°58.02	2	
1,200	113	1,222	37°15.06	176°57.98	1	
1,500	105	1,486	37°11.35	176°56.59	3	
White island	700	154	700	37°37.05	177°13.46	1	
700	155	704	37°37.04	177°13.48	2	
1,000	150	1,017	37°33.14	177°16.21	1	
1,000	151	1,023	37°33.20	177°16.10	1	
1,000	152	1,031	37°33.17	177°16.05	1	
1,200	142	1,200	37°31.75	177°17.71	1	
1,200	143	1,202	37°31.77	177°17.69	2	
1,500	135	1,523	37°26.59	177°21.05	3	
Runaway	700	55	705	37°25.85	177°53.62	3	
1,000	60	900	37°24.17	177°52.65	2	
1,000	61	870	37°24.20	177°52.67	1	
1,200	65	1,254	37°21.86	177°52.59	1	
1,200	66	1,254	37°21.86	177°52.59	2	
1,500	70	1,518	37°18.13	177°52.27	3	
Note:

N, number of cores.

Meiofauna samples from seep habitats in the Hikurangi Margin at two sites geographically close to the other habitats sites were obtained from a previous survey in 2006 (voyage TAN0616) (Table 1), and were used in a second-stage analysis comparing seep, canyon, seamount, and slope communities (see below).

Macrofaunal and mega-epifaunal communities were also sampled using towed cameras, corers, trawls, and epibenthic sled at the same depth strata at each habitat during the two main surveys, and the results of the analyses of data for these components of the benthos have and will be reported elsewhere (Bowden et al., 2016; Leduc et al., in press). Data on meiofauna are reported here for the first time.

Sampling and sample processing

Meiofauna and sediment samples were collected using an Ocean Instruments MC-800A multicorer (internal diameter core = 9.52 cm). At each station, one to three cores were used from each multicorer deployment for meiofaunal samples (refer Table 1), and one core for a sediment sample. Each meiofaunal core was sliced into three vertical fractions: 0–1, 1–3 and 3–5 cm sediment depth layers and preserved in 10% buffered formalin. Previous analysis showed there was small difference between 1–3 and 3–5 cm layers, therefore these layers were combined prior to sieving. Samples were rinsed on a 1 mm mesh sieve to remove macrofauna and on a 45 μm mesh to retain meiofauna. Meiofauna were extracted from the sieved sediment by Ludox flotation (Somerfield & Warwick, 1996) and were identified to main taxa (e.g., nematodes, nauplii, copepods, annelids) (Higgins & Thiel, 1988) under a stereomicroscope.

The following physical and biogeochemical parameters were determined from the sediment samples: mean particle size (geometric), sorting, skewness, kurtosis, %silt/clay, particle size diversity (PSD; calculated using Shannon-Wiener diversity index of 11 particle size classes (after Etter & Grassle, 1992)), calcium carbonate content (%CaCO3), organic matter content (%OM), organic carbon content (%OC), nitrogen content (%N), chlorophyll a concentration (chl a) and phaeopigment concentration (phaeo) using methods described by Grove et al. (2006), Nodder et al. (2007) and Nodder et al. (2003). The %CaCO3 was determined from the top 5 cm of sediment, whereas organic matter (%OM, %OC and %N) was determined from the top 1 cm of sediment.

Additional environmental characterisation

The environmental data used in the present study were first published in Bowden et al. (2016). Surface water chlorophyll concentrations were determined using ocean colour estimates of surface chlorophyll concentrations as a proxy for long-term inter-station variability in primary production (NASA SeaWiFS Project: http://oceandata.sci.gsfc.nasa.gov/SeaWiFS/Mapped/8Day/9km/chlor_a). The 9 km composited data of surface chlorophyll were further composited to 90 × 90 km pixels centred on the location of each sample station. The mean value for the 1997–2010 period was computed for each station.

Seafloor habitats at the study sites were characterised using seafloor morphology derivatives from MBES data gridded at 25 m resolution. The following topographic variables were derived for each sampling station: depth, slope (steepest gradient to any neighbouring cell), curvature (change of slope), plan curvature (curvature of the surface perpendicular to the slope direction), and profile curvature (curvature of the surface in the direction of slope). A further set of derivatives was calculated for the standard deviation of depth, depth range, standard deviation of the slope (a proxy measure for slope roughness), and terrain rugosity based on a 3, 5, 7, and 15 grid cell focal means. A total of 18 topographic variables were used in the analysis. Methods for the determined topographic variables are provided by Nodder et al. (2013).

Trawl effort data were used to quantify the extent of commercial fishing intensity conducted on the seafloor in the study regions. Estimates of fishing intensity for a 5 × 5 km cell grid covering the New Zealand Exclusive Economic Zone were derived using the number of tows and an estimate of swept area derived from the trawl width and either the distance between start and finish positions, or the tow duration (Black & Wood, 2014). Fishing intensity at each of the study stations was estimated for the total trawled area within the corresponding 5 × 5 km cell integrated over a period of ten years prior to sampling.

Statistical analysis

Statistical analyses were conducted to test the following main hypotheses: that there is no difference in meiofaunal community attributes (abundance, diversity and community structure) in surface (0–1 cm) and subsurface (1–5 cm) sediment layers among deep-sea habitats, water depths, and between regions, and that there is no relationship between bottom trawling or environmental variables and meiofaunal community attributes.

Analyses of meiofaunal community attributes (abundance, diversity, and community structure) were conducted using statistical routines in the multivariate software package PRIMER v6 with PERMANOVA (Anderson, Gorley & Clarke, 2008; Clarke & Gorley, 2006). Meiofaunal taxon richness was used as the measure of meiofaunal diversity. All analyses were conducted on individual core data.

Analysis of community structure was based on fourth-root transformed abundance data (abundance data per core at each station). Fourth-root transformation was used to reduce contributions to similarity by the numerically dominant nematodes (Somerfield & Clarke, 1995). Similarity matrices for the community structure analysis were built using Bray-Curtis similarity (Clarke & Gorley, 2006). Similarity matrices for meiofauna abundance and diversity were based on Euclidean distance similarity matrices of untransformed data.

The PERMANOVA routine in PRIMER was used to investigate the relative influences of survey region, habitat, water depth strata and sediment depth on community attributes (Anderson, Gorley & Clarke, 2008). Preliminary analysis showed a significant difference in the abundance of meiofauna between the two regions. Therefore, in addition to a single-factor test for the effect of region (Hikurangi Margin versus Bay of Plenty), and to avoid an overriding influence of abundance on patterns of community structure, analysis testing for the effects of habitat, water depth, and sediment depth were conducted for each region separately. Data were analysed using a four-factor design, with the factors habitat (fixed; canyon, seamount, slope), water depth (fixed; 700, 1,000, 1,200, 1,500 m), sediment depth (fixed; 0–1 and 1–5 cm), and cores (random, nested within habitat and water depth strata). P-values for individual predictor variables were obtained using 9,999 permutations. Lack of independence between stations due to geographical proximity (i.e., spatial autocorrelation/structure) is common in natural communities and poses limitations for the interpretation of ecological patterns (Legendre, 1993). In particular, failure to take into account the spatial component of ecological variation may affect tests of statistical significance when investigating relationships between community structure and environmental parameters (Legendre & Troussellier, 1988). Therefore, latitude and longitude were fitted first in the models of community structure to account for the effect of geographical proximity. The main factor test was followed by pair-wise tests when significant effects were found. The square-root value of estimates of components of variation (√ECV) was used to compare the relative strengths of significant factor effects. A non-metric multi-dimensional scaling plot (MDS) was used to visualise patterns in multivariate community structure. The SIMPER routine was used to identify which taxa were responsible for any habitat, region, water depth strata and/or sediment-related differences in community structure. SIMPER was also used to identify the main environmental variables responsible for differences between regions. This analysis was conducted on similarity matrices built using normalised environmental data and the Euclidean similarity measure; topographical variables were not included because slope, canyon, and seamount habitats are defined a priori as topographical features.

The DistLM routine was used to investigate the relationship between meiofaunal community attributes and environmental variables. The full set of environmental variables was partitioned into five sets, i.e., spatial (water depth), sediment characteristics (mean particle size, sorting, skewness, kurtosis, %silt/clay, PSD, %CaCO3, %OM, %N, %OC, chl a, phaeopigment), primary productivity (surface chlorophyll concentration), fishing intensity, and topography variables (18 variables). Environmental variables that were strongly correlated (r > 0.8) were removed prior to analysis (Table S1). Relationships between environmental parameters and community attributes were initially examined by analysing each predictor separately (marginal tests). Partial regressions were used to better characterise the relationships and to account for the effect of the remaining variables. Sequential tests were conducted using step-wise selection procedures and R2 as the selection criterion. Latitude and longitude were fitted first in the models of community structure to account for the effect of geographical proximity. P-values for individual predictor variables were obtained using 9,999 permutations.

Meiofaunal community data from slope, canyon, and seamount habitats in Hikurangi Margin (from TAN1004) were compared to those for seep meiofauna in a second-stage analysis. Stations from the two seep sites were available from 1,049–1,059 m water depths (Table 1), thus only data from the depth strata of, and closest in depth to, these sites were included in the analysis (i.e., the 1,500 m stratum was excluded). The effects of habitat and sediment depth on meiofaunal community attributes were compared using PERMANOVA. MDS and SIMPER routines were conducted as described above.

Results

In total, 15 meiofaunal taxa were identified from the samples. The most abundant taxon was nematodes (87.1% of total abundance), followed by copepods (6.0%), nauplii (4.2%) and annelids (1.4%). The abundance of each of the remaining taxa (e.g., ostracods, kinorhynchs, isopods, tanaidaceans, amphipods, gastrotrichs, loriciferans, tardigrades, bivalves, cumaceans, aplacophorans) was less than 0.8% of total meiofaunal abundance.

Comparison of Hikurangi Margin and Bay of Plenty regions

SIMPER analysis of environmental variables showed substantial variability between regions, mostly in surface water chlorophyll concentration, sediment phaeopigment concentration, organic carbon content of the sediment and fishing intensity (Table S2). These four variables were substantially higher in the Hikurangi Margin than in the Bay of Plenty (Fig. 2). Surface water chlorophyll concentrations and organic carbon content were two times higher, and sediment phaeopigment concentration five times higher, in the Hikurangi Margin than in the Bay of Plenty. Mean fishing intensity was 30 times greater in the former region, but among-site variability was high.

Figure 2 Comparison of variables responsible for most of environmental dissimilarity between the Hikurangi Margin and Bay of Plenty study regions.

(A) Mean surface chlorophyll concentration; (B) Sediment phaeopigment concentration; (C) Sediment organic carbon content; (D) Fishing intensity (Environmental data first published in Bowden et al. (2016)).

There was a significant difference in meiofaunal abundance between regions (PERMANOVA, P = 0.0001). Average meiofaunal abundance was higher in the Hikurangi Margin (1,481 ± 538 individual 10 cm−2) compared to the Bay of Plenty (929 ± 396). There was a small but significant difference in meiofaunal diversity (meiofaunal taxon richness) between regions (PERMANOVA, P = 0.04), with a total of 12 major taxa identified in the Hikurangi Margin region (average diversity: 7 taxa per core), which was less than the 14 major taxa identified in the Bay of Plenty region (average diversity: 6 taxa per core).

Meiofaunal community structure was significantly different between regions (PERMANOVA, P = 0.0001; Fig. 3). Kinorhynchs were the greatest contributor to between-region dissimilarity, and like most other meiofaunal taxa, their average abundance was higher in the Hikurangi Margin region than in the Bay of Plenty (Table 2). Tardigrades, gastrotrichs and loriciferans were only recorded in the Bay of Plenty, whereas bivalves were only recorded in the Hikurangi Margin.

Figure 3 Two-dimensional MDS ordination plot of meiofaunal community structure at the Hikurangi Margin and Bay of Plenty study regions.

Table 2 SIMPER analysis results showing meiofaunal taxa accounting for community dissimilarity between the Hikurangi Margin and Bay of Plenty study regions (cut-off applied at 90% contribution).

Taxon	Av.abund	Av.abund	Av.Diss	Diss/SD	Contrib%	Cum.%	
	Hikurangi margin	Bay of Plenty					
Kinorhynchs	6.6	1.5	2.96	1.31	13.25	13.25	
Ostracods	2.9	0.9	2.66	1.25	11.89	25.15	
Nematodes	663.4	446.5	2.32	1.5	10.39	35.54	
Copepods	58.1	21.0	2.25	1.54	10.06	45.6	
Tanaidaceans	1.1	0.4	2.18	1.11	9.75	55.36	
Nauplii	38.4	16.6	2.18	1.39	9.74	65.1	
Tardigrades	0	1.0	1.95	0.95	8.72	73.82	
Annelids	14.0	4.4	1.84	1.53	8.25	82.06	
Isopods	0.50	0.3	1.34	0.74	6.01	88.08	
Amphipods	0.40	0.2	1.19	0.69	5.31	93.38	
Note:

[Av.abund, average meiofauna abundance (individual 10 cm−2); Av.Diss, average dissimilarity; Diss/SD, Dissimilarity/Standard Deviation; Contrib%, % contribution to overall dissimilarity; Cum.%, % cumulative dissimilarity]. Higher average abundance are shown in bold.

Hikurangi Margin

Meiofaunal abundance differed significantly among habitats, water depths, and sediment depths in the Hikurangi Margin study region (PERMANOVA, P < 0.05; Fig. 4; Table S3). Interactions between sediment depth and all the other factors were also significant, indicating that patterns were not consistent between surface and subsurface layers. Pairwise comparisons showed significantly lower abundance of surface (0–1 cm) meiofauna on seamounts relative to canyons, while subsurface (1–5 cm) meiofaunal abundance was significantly lower on seamounts than in both canyon or slope habitats. Pairwise comparisons also showed significantly higher abundance of surface and subsurface meiofauna at 700 m water depth than deeper depths (surface layer: 1,200 and 1,500 m, subsurface layer: 1,000, 1,200 and 1,500 m). Comparing the estimates of components of variation showed that sediment depth (89.0) and habitat (86.8) explained similar proportions of variability in abundance, whilst water depth explained a smaller proportion (57.4) (Table S3). Diversity differed significantly between sediment depths, but not among habitats or water depths (PERMANOVA, P < 0.05; Table S4), and higher in surface than in subsurface sediments.

Figure 4 Comparison of average total meiofaunal abundance among habitats (slope, canyon and seamount) in Hikurangi Margin sand Bay of Plenty.

Data are means (± SD). nd, no data.

Meiofaunal community structure differed significantly between sediment depths, but not among habitats or water depths (PERMANOVA, P = 0.0001; Fig. 5; Table S5). SIMPER analysis showed average community dissimilarity between the 0–1 and 1–5 cm sediment depth was 24.2%; nauplii were the largest contributor to community dissimilarity (16.8% of total dissimilarity) (Table 3). Average abundance of nematodes, annelids, and isopods was higher in the 1–5 cm than the 0–1 cm sediment depth layer, whereas the other meiofaunal taxa showed the opposite trend.

Figure 5 Two-dimensional MDS ordination of meiofaunal community structure at the study regions.

Hikurangi Margin: (A) Habitat; (B) Water depth; (C) Sediment depth, Bay of Plenty; (D) Habitat; (E) Water depth; (F) Sediment depth.

Table 3 SIMPER analysis results showing meiofaunal taxa accounting for community dissimilarity between 0–1 and 1–5 cm sediment depth layers in the Hikurangi Margin study region (cut-off applied at 90% contribution).

Taxon	Av.abund	Av.abund	Av.Diss	Diss/SD	Contrib%	Cum.%	
	0–1 cm	1–5 cm					
Nauplii	28.7	9.7	4.06	1.23	16.78	16.78	
Kinorhynchs	4.3	2.3	3.35	1.2	13.85	30.63	
Copepods	39.1	19.0	3.21	1.08	13.26	43.89	
Ostracods	2.2	0.7	3.07	1.17	12.67	56.56	
Nematodes	248.3	415.2	3.02	1.34	12.48	69.05	
Tanaidaceans	0.7	0.8	2.13	0.93	8.81	77.86	
Annelids	4.9	9.1	2.09	1.03	8.62	86.48	
Isopods	0.2	0.3	1.1	0.58	4.52	91	
Note:

[Av.abund, average meiofauna abundance (individual 10 cm−2); Av.Diss, average dissimilarity; Diss/SD, Dissimilarity/Standard Deviation; Contrib%, % contribution to overall dissimilarity; Cum.%, % cumulative dissimilarity]. Higher average abundance are shown in bold.

Results of DistLM analyses showed that abundance in the 0–1 cm sediment layer was significantly correlated with profile curvature and water depth (P < 0.05; R2 = 0.12; Table 4; Fig. 6). Abundance in surface sediment was negatively correlated with profile curvature, indicating that abundance was greater in depressions than on elevated topography, whereas the relationship between abundance and water depth was positive. Meiofaunal abundance in the 1–5 cm layer was significantly and positively correlated with the standard deviation of the slope (15 grid cell focal mean; a proxy measure for slope roughness), and sediment phaeopigment concentration (P < 0.05; R2 = 0.24–0.41). Abundance in subsurface sediment was also negatively correlated with water depth (P < 0.05; R2 = 0.19; Table 4).

Table 4 DistLM analysis results showing correlations between environmental variables and meiofaunal attributes for the Hikurangi Margin region.

Variable	P	R2	Variable	P	R2	R2 cum	rs.df	
Marginal tests			Sequential tests					
Abundance 0–1 cm	Abundance 0–1 cm	
(−) Profile curvature	0.0073	0.12	(−) Profile curvature	0.0075	0.12	0.12	57	
(+) Water depth	0.0074	0.12	(+) Water depth	0.0087	0.10	0.22	56	
(−) Curvature	0.0124	0.11	(−) %CaCO3	0.0295	0.06	0.28	55	
Abundance 1–5 cm	Abundance 1–5 cm	
(+) Slope STD15	0.0001	0.41	(+) Slope STD15	0.0001	0.41	0.41	57	
(+) Vrm05	0.0012	0.25	(−) %CaCO3	0.0007	0.12	0.53	56	
(+) Phaeopigment	0.0003	0.24	(+) Vrm05	0.0013	0.11	0.64	55	
(−) Water depth	0.0006	0.19	(+) Slope STD03	0.0005	0.07	0.71	54	
(+) Skewness	0.0022	0.14						
(−) Curvature	0.0209	0.09						
(+) STD15	0.0363	0.07						
(+) Particle size diversity	0.0475	0.07						
Diversity 0–1 cm	Diversity 0–1 cm	
(−) Mean particle size	0.0407	0.07	(−) Mean particle size	0.0418	0.07	0.07	57	
(−) Particle size diversity	0.0444	0.07						
(−) Fishing intensity	0.0445	0.06						
Diversity 1–5 cm	Diversity 1–5 cm	
(−) Curvature	0.0010	0.16	(−) Curvature	0.0019	0.16	0.16	57	
(−) Profile curvature	0.0039	0.14						
(+) Phaeopigment	0.0068	0.13						
(−) %CaCO3	0.0073	0.12						
Community structure 0–1 cm	Community structure 0–1 cm	
Profile curvature	0.0008	0.08	Profile curvature	0.0008	0.08	0.08	57	
Curvature	0.0017	0.07	%CaCO3	0.0057	0.05	0.18	55	
Depth	0.0145	0.05	Depth	0.0207	0.04	0.12	56	
Skewness	0.0227	0.05	Vrm05	0.0293	0.03	0.24	53	
%CaCO3	0.0466	0.04						
Community structure 1–5 cm	Community structure 1–5 cm	
Curvature	0.0020	0.09	Curvature	0.0021	0.09	0.09	57	
Phaeopigment	0.0023	0.08	Slope STD15	0.0025	0.08	0.16	56	
Slope STD15	0.0022	0.08	%CaCO3	0.0074	0.05	0.21	55	
%CaCO3	0.0106	0.06						
Profile curvature	0.0123	0.06						
Vrm05	0.0131	0.06						
Depth	0.0232	0.05						
Note:

[P, probability; R2, proportion of explained variation attributable to each variable; R2 (cum), cumulative proportion of variation; rs.df, residual degrees of freedom; Slope STD, Standard deviation of slope based on 3, 5, 7, 15 grid cell focal mean; STD, Standard deviation of depth; Vrm, terrain rugosity; (+/−), positive/negative relationship].

Figure 6 Selection of statistically significant (P < 0.05) correlations between environmental variables and meiofaunal abundance at different sediment layers in the Hikurangi Margin and Bay of Plenty regions.

Hikurangi Margin, surface sediment (0–1 cm): (A) Profile curvature; (B) Water depth (m); Subsurface sediment (1–5 cm): (C) Standard deviation of the slope (15 grid cell focal mean); (D) Phaeopigment concentration (μg/g); Bay of Plenty, surface sediment (0–1 cm): (E) Surface chlorophyll concentration (mg m−3); (F) Plan curvature; Subsurface sediment (1–5 cm): (G) Organic carbon content (%OC); (H) Kurtosis. (See Tables 4 and 6 for results of DistLM analyses).

Meiofaunal diversity in the 0–1 cm sediment layer was significantly and negatively correlated with mean particle size, particle size diversity and fishing intensity (P < 0.05; R2 = 0.07; Table 4; Fig. 7). Diversity in the 1–5 cm sediment layer was significantly and negatively correlated with both curvature and profile curvature (P < 0.05; R2 = 0.13–0.16), indicating that diversity was greater in depressions than on elevated topography.

Figure 7 Selection of statistically significant (P < 0.05) correlations between environmental variables and meiofaunal diversity at different sediment layers in the Hikurangi Margin and Bay of Plenty study regions.

Hikurangi Margin, surface sediment (0–1 cm): (A) Particle size diversity; (B) Fishing intensity (num. of trawls); Subsurface sediment (1–5 cm): (C) Curvature; (D) Profile curvature; Bay of Plenty, surface sediment (0–1 cm): (E) Surface chlorophyll concentration (mg m−3); (F) Water depth (m); Subsurface sediment (1–5 cm): (G) Kurtosis; (H) Organic carbon content. (See Tables 4 and 6 for results of DistLM analyses).

Meiofaunal community structure in the 0–1 cm sediment layer was significantly correlated with profile curvature, curvature and water depth (P < 0.05; R2 = 0.05–0.08; Table 4), whilst community structure in the 1–5 cm sediment layer was significantly correlated with curvature and phaeopigment concentration in the sediment.

Bay of Plenty

Meiofaunal abundance differed significantly among habitats, water depth, and sediment depths in the Bay of Plenty study region; there was also a significant interaction between habitat and water depth (PERMANOVA, P < 0.05; Table S3; Fig. 4). Pairwise comparisons only showed a significant interaction at 1,200 m, but not at other water depths, where higher abundance of meiofauna were observed in canyons relative to slopes. Comparing the estimates of components of variation showed that habitat explained a greater proportion of the variability in abundance than sediment depth and water depth (Table S3). Diversity differed significantly between sediment depth, but not among habitats or water depths (PERMANOVA, P < 0.05; Table S4), and higher in surface than in subsurface sediments.

Meiofaunal community structure differed significantly among water depths and between sediment depths, but not among habitats (PERMANOVA, P < 0.05; Fig. 5; Table S5). Comparing the estimates of components of variation showed that sediment depth explained a greater proportion of the variability in abundance than water depth (Table S5). Pairwise comparisons showed that community structure differed significantly between 700 and 1,200 m, and between 700 and 1,500 m. SIMPER analysis showed average community dissimilarity between 700 and 1,200 m, and between 700 and 1,500 m depth, was ∼ 24%. SIMPER results showed that nauplii were the main contributor to community dissimilarity, and that the average abundance of all meiofaunal taxa was higher at 700 m than at 1,200 and 1,500 m water depths (Table 5). Average community dissimilarity between 0–1 and 1–5 cm sediment depths was 26.3%. SIMPER results showed that nauplii were the main contributor to community dissimilarity, and that nematode average abundance was higher in the 1–5 cm than the 0–1 cm sediment depth, whereas the other meiofaunal taxa showed the opposite trend (Table 5).

Table 5 SIMPER analysis results showing meiofaunal taxa accounting for community dissimilarity between different water and sediment depths for the Bay of Plenty study region (cut-off applied at 70% contribution).

Taxon	Av.abund	Av.abund	Av.Diss	Diss/SD	Contrib%	Cum.%	
Water depths	700 m	1,200 m					
Nauplii	24.4	12.0	4.05	1.17	17.02	17.02	
Annelids	5.8	3.2	3.19	1.08	13.4	30.42	
Kinorhynchs	1.7	1.2	2.78	0.94	11.65	42.07	
Tardigrades	1.6	0.6	2.57	0.89	10.81	52.89	
Nematodes	441.6	385.1	2.52	1.31	10.59	63.48	
Copepods	28.0	19.4	2.47	1.38	10.36	73.84	
	700 m	1,500 m					
Nauplii	24.4	14.1	3.94	1.17	16.52	16.52	
Copepods	28.0	16.4	3.35	1.03	14.04	30.57	
Kinorhynchs	1.7	1.5	2.88	0.97	12.08	42.65	
Nematodes	441.6	431.3	2.73	1.31	11.44	54.08	
Tardigrades	1.6	0.9	2.72	0.95	11.41	65.49	
Ostracods	1.6	0.7	2.3	0.83	9.64	75.13	
Sediment depths	0–1 cm	1–5 cm					
Nauplii	13.6	3.0	5.06	1.34	19.24	19.24	
Copepods	15.2	5.8	3.28	1.23	12.49	31.73	
Kinorhynchs	1.2	0.3	3.23	1.03	12.27	44	
Nematodes	184.9	261.6	2.99	1.3	11.39	55.4	
Tardigrades	0.8	0.2	2.74	0.94	10.42	65.82	
Annelids	2.4	2.1	2.51	0.9	9.53	75.35	
Note:

[Av.abund, average meiofauna abundance (individual 10 cm−2); Av.Diss, average dissimilarity; Diss/SD, Dissimilarity/Standard Deviation; Contrib%, % contribution to overall dissimilarity; Cum.%, % cumulative dissimilarity]. Higher average abundance are shown in bold.

Results of DistLM analysis showed that abundance in the 0–1 cm sediment layer was significantly correlated with surface water chlorophyll concentration, sediment carbonate content and plan curvature (P < 0.05; R2 = 0.09; Table 6). The relationship between abundance and sediment carbonate content was negative, whereas abundance was positively correlated with surface water chlorophyll concentration and plan curvature (Fig. 6). The positive relationship between plan curvature and abundance indicated that abundance was greater in elevated topography perpendicular to the slope direction. Abundance in the 1–5 cm sediment layer was significantly and positively correlated with organic carbon content, kurtosis, and phaeopigment concentration in the sediment (P < 0.05; R2 = 0.07–0.18; Table 6; Fig. 6).

Table 6 DistLM analysis results showing correlations between environmental variables and meiofaunal community attributes in the Bay of Plenty region.

Variable	P	R2	Variable	P	R2	R2 cum	rs.df	
Marginal tests			Sequential tests					
Abundance 0–1 cm	Abundance 0–1 cm	
(+) Surface chlorophyll concentration	0.0125	0.09	(+) Surface chlorophyll concentration	0.0119	0.09	0.09	69	
(−) %CaCO3	0.0140	0.09	(+) Plan curvature	0.0172	0.09	0.17	68	
			(+) %OC	0.0170	0.07	0.24	67	
			(−) Skewness	0.0356	0.05	0.29	66	
			(+) Slope STD07	0.0265	0.05	0.40	62	
Abundance 1–5 cm	Abundance 1–5 cm	
(+) %OC	0.0001	0.18	(+) %OC	0.0005	0.18	0.18	69	
(+) Kurtosis	0.0012	0.14	(+) Kurtosis	0.0115	0.07	0.26	68	
(+) Phaeopigment	0.0057	0.10	(+) Range07	0.0212	0.05	0.53	53	
(+) %Silt/clay	0.0141	0.08						
(−) Sorting	0.0163	0.08						
Diversity 0–1 cm	Diversity 0–1 cm	
(+) Surface chlorophyll concentration	0.0053	0.10	(+) Surface chlorophyll concentration	0.0057	0.10	0.10	69	
(−) Water depth	0.0107	0.09	(−) Sorting	0.0244	0.07	0.17	68	
(+) Phaeopigment	0.0100	0.09						
Diversity 1–5 cm	Diversity 1–5 cm	
(+) Kurtosis	0.0014	0.15	(+) Kurtosis	0.0013	0.15	0.15	69	
(+) %OC	0.0106	0.09						
(+) Skewness	0.0220	0.07						
(+) %Silt/clay	0.0335	0.06						
(+) %OM	0.0369	0.06						
Community structure 0–1 cm	Community structure 0–1 cm	
Water depth	0.0056	0.04	Water depth	0.0062	0.04	0.04	69	
Surface chlorophyll concentration	0.0110	0.04	Plan curvature	0.0082	0.03	0.07	68	
Plan curvature	0.0174	0.03	%OM	0.0220	0.03	0.14	66	
%Silt/clay	0.0294	0.03	Kurtosis	0.0302	0.03	0.28	59	
Phaeopigment	0.0306	0.03						
Sorting	0.0418	0.03						
%OM	0.0403	0.03						
Community structure 1–5 cm	Community structure 1–5 cm	
Kurtosis	0.0003	0.06	Kurtosis	0.0004	0.06	0.06	69	
%Silt/clay	0.0030	0.05	%OC	0.0448	0.03	0.09	68	
Skewness	0.0048	0.05	Surface chlorophyll concentration	0.0475	0.03	0.16	65	
%OC	0.0091	0.04						
Sorting	0.0252	0.03						
%OM	0.0276	0.03						
Phaeopigment	0.0274	0.03						
Note:

[P, probability; R2, proportion of explained variation attributable to each variable; R2 (cum), cumulative proportion of variation; rs.df, residual degrees of freedom; Slope STD, Standard deviation of slope based on 3, 5, 7, 15 grid cell focal mean; STD, Standard deviation of depth; (+/−), positive/negative relationship].

Meiofaunal diversity in the 0–1 cm sediment layer was significantly correlated with surface water chlorophyll concentration and water depth (P < 0.05; R2 = 0.09–0.1; Table 6; Fig. 7). The relationship between diversity and surface water chlorophyll concentration was positive, whereas diversity was negatively correlated with water depth. Diversity in the 1–5 cm sediment layer was significantly and positively correlated with kurtosis and organic carbon content (P < 0.05; R2 = 0.09–0.15).

Meiofaunal community structure in the 0–1 cm sediment layer was significantly correlated with water depth and surface water chlorophyll concentration (P < 0.05; R2 = 0.04; Table 6). Community structure in the 1–5 cm sediment layer was significantly correlated with kurtosis, silt and clay particle content, and particle skewness (P < 0.05; R2 = 0.05–0.06).

Slope, canyon, and seamount habitats compared to seep habitat: Hikurangi Margin

The second-stage analysis of slope, canyon, seamount, and seep communities in the Hikurangi Margin showed a significant effect of habitat, sediment depth, and their interaction, on abundance (PERMANOVA, P < 0.05). Pairwise comparisons only showed a significant interaction between canyons and seeps at subsurface sediment (1–5 cm), where abundance was higher in canyon than seep habitats (Table S6). Diversity differed significantly among habitats and between sediment depths (PERMANOVA, P < 0.05). Differences in diversity were small, but overall diversity was significantly higher in seep habitat (average diversity = 7.2) compared to the other habitats (canyon = 5.9, slope = 6.0, seamount = 5.4), and was significantly higher in surface sediment (6.6) than in subsurface sediment (5.7).

Meiofaunal community structure differed significantly among habitats and between sediment depths (PERMANOVA, P < 0.05; Table S7). Pairwise comparisons showed that meiofaunal communities differed significantly (P < 0.05) between seep and all of the other habitats, which did not differ significantly from each other (Fig. 8). Nauplii and amphipods contributed the most to community dissimilarity (12–15% of total dissimilarity) between seeps and the other habitats (Table 7). Average abundance of meiofaunal taxa was higher in seep habitats than in the other habitats, except for kinorhynchs, ostracods and nematodes which were most abundant in canyon and slope habitats.

Figure 8 Two-dimensional MDS ordination of meiofaunal community structure for habitats in the Hikurangi Margin study region (water depth: 700–1,200 m only).

Depth strata are shown by shades of grey ranging from light grey (700 m) to black (1,200 m).

Table 7 Results of the SIMPER analysis showing meiofauna taxa accounting for community dissimilarity between seep and other habitats for the Hikurangi Margin study region (cut-off applied at 90% contribution).

Taxon	Av.abund	Av.abund	Av.Diss	Diss/SD	Contrib%	Cum.%	
	Seamount	Seep					
Nauplii	8.6	30.3	4.04	1.19	14.73	14.7	
Amphipods	0.2	3.6	3.59	1.32	13.07	27.8	
Copepods	16.9	36.8	3.3	1.13	12.01	39.8	
Tanaidaceans	0.3	2.5	3.2	1.27	11.65	51.5	
Kinorhynchs	2.4	2.6	3.06	1.15	11.15	62.6	
Ostracods	0.6	1.5	2.74	1.13	9.98	72.6	
Bivalves	0	1.0	2.46	1.04	8.97	81.6	
Nematodes	213.4	268.2	2.36	1.39	8.6	90.2	
	Canyon	Seep					
Nauplii	19.8	30.3	3.8	1.21	14.28	14.3	
Amphipods	0.2	3.6	3.4	1.35	12.76	27.0	
Kinorhynchs	4.3	2.6	3.07	1.17	11.54	38.6	
Tanaidaceans	0.5	2.5	2.87	1.24	10.8	49.4	
Copepods	30.3	36.8	2.87	1.27	10.8	60.2	
Ostracods	1.8	1.5	2.71	1.16	10.19	70.4	
Nematodes	398.5	268.2	2.69	1.31	10.1	80.5	
Bivalves	0.4	1.0	2.32	1.06	8.74	89.2	
Annelids	9.1	9.3	1.66	1.11	6.25	95.5	
	Slope	Seep					
Amphipods	0.2	3.6	3.45	1.36	13.15	13.2	
Nauplii	22.6	30.3	3.32	1.24	12.68	25.8	
Tanaidaceans	0.5	2.5	3.01	1.28	11.47	37.3	
Kinorhynchs	3.6	2.6	2.89	1.14	11.04	48.4	
Copepods	34.9	36.8	2.89	1.17	11.01	59.4	
Ostracods	1.7	1.5	2.65	1.16	10.1	69.5	
Nematodes	315.5	268.2	2.38	1.27	9.07	78.5	
Bivalves	0.03	1.0	2.28	1.04	8.7	87.2	
Annelids	5.3	9.3	1.83	0.83	6.99	94.2	
Note:

[Av.abund, average meiofauna abundance (individual 10 cm−2); Av.Diss, average dissimilarity; Diss/SD, Dissimilarity/Standard Deviation; Contrib%, % contribution to overall dissimilarity; Cum.%, % cumulative dissimilarity]. Higher average abundance are shown in bold.

Discussion

Knowledge of the benthic communities associated with distinct habitats in the deep sea has increased significantly during the last decades, as we now have a better understanding of how substrate type and availability, biogeochemistry, nutrient input, productivity, hydrographic conditions and catastrophic events shape community patterns on regional scales (Levin et al., 2010; Vanreusel et al., 2010). In this study, meiofaunal community attributes differed between regions and sediment depths, and between habitats and water depths for some community attributes. Relationships between environmental variables, trawling intensity, and community attributes also differed between surface and subsurface sediment communities. The patterns observed are discussed below in relation to potential environmental drivers, as is the relative vulnerability of meiofaunal communities to anthropogenic activities.

Regional differences in meiofaunal communities

The flux of organic matter from the surface to the seafloor is the main driver of meiofaunal benthic abundance (Lambshead et al., 2002; Soltwedel, 2000). Meiofaunal density has often been linked to food availability in the sediment (Ingels et al., 2009; Lampadariou & Tselepides, 2006; Leduc et al., 2014), with high food concentrations associated with high numbers of individuals. The greater abundance of meiofauna in the Hikurangi Margin relative to the Bay of Plenty appeared to be related to differences in surface water chlorophyll concentrations. The latter corresponded with phaeopigment concentrations and organic carbon content of the sediment, indicating increased food availability in the Hikurangi Margin than in the Bay of Plenty region. It is likely that this higher food availability led to the observed differences in meiofaunal abundance. There was a clear difference in meiofaunal community structure between regions. Most taxa were more abundant in the Hikurangi Margin compared to the Bay of Plenty, except for certain rare taxa (e.g., gastrotrichs, tardigrades, loriciferans, bivalves) that were only present in one of the regions.

Trawling activity can have pronounced effects on meiofaunal communities (Pusceddu et al., 2014; Schratzberger et al., 2009), and could also be responsible for regional differences in community attributes in the present study. Although there was only a weak correlation between trawling intensity and diversity in surface sediments, it is possible that trawling impacts on environmental variables may have affected diversity. For example, trawling has been shown to alter sediment physical characteristics and the distribution of organic matter in the sediment column, through continuous stirring of the upper sediments which leads to removal of recent organic-rich sediment and induced changes in the grain size distribution, as repeated resuspension of the remaining sediments favours the sorting of particles according to their settling speeds (Martín et al., 2014; Pusceddu et al., 2014). In the present study, we found a negative relationship between mean particle size and particle size diversity and meiofaunal diversity in the surface sediment of the Hikurangi Margin, which contrasts with the findings of previous studies showing the opposite pattern (Etter & Grassle, 1992; Leduc et al., 2012). This discrepancy may be explained by the impacts of trawling, which could increase mean sediment particle size and sediment particle size diversity while at the same time decreasing diversity through increased dominance of opportunistic genera (Pusceddu et al., 2014; Schratzberger et al., 2009). However, identifying potential impacts of trawling at the regional scale will require further research.

Among-habitat differences in meiofaunal communities

Meiofaunal abundance differed among the deep-sea habitats studied, which was evident in both of the study regions. The first-stage analysis showed that abundance was higher in canyons than in other habitats of both regions. Abundance also differed between water depths in both study regions, with total meiofaunal abundance consistently higher in the shallower strata.

In the Hikurangi Margin region, profile curvature and water depth were the two factors most strongly correlated with abundance in the surface sediment. Greater meiofaunal abundance in seafloor depressions could be associated with greater settlement of meiofauna associated with slower near-bottom water currents in depressions (Fleeger, Yund & Sun, 1995; Giere, 2009). Negative profile curvatures were mostly found in canyon habitat (see Fig. 6), which is well known for their complex topography (Canals et al., 2006), and could partly explain the observed canyon habitat effect. In addition, abundances for surface sediments were positively correlated with water depth in all habitats. Higher abundance at deeper sites could result from high settlement of meiofauna that was passively transported downslope by currents; even weak currents can re-suspend meiofaunal organisms and transport them long distances down continental margins (Boeckner, Sharma & Proctor, 2009; Pusceddu et al., 2014). Higher abundance could also be related to increase in food availability at deeper depths observed in this study, which may result from downslope transport of fine organic matter (Pusceddu et al., 2014; Weaver et al., 2000).

In the Bay of Plenty region, other environmental variables influenced meiofaunal abundance. In the surface sediment, surface chlorophyll concentration and plan curvature were positively correlated with abundance. Surface water chlorophyll concentration can be considered an indicator of the flux of organic matter and phytodetritus to the sea floor, and thereby the availability of food to benthic organisms (Rex & Etter, 2010). In the present study, surface water chlorophyll concentrations corresponded with higher meiofaunal abundance at the canyon sites, and previous studies support this finding (Baguley et al., 2006; Ingels et al., 2009; Pusceddu et al., 2009; Soltwedel, 2000). The positive relationship between plan curvature and abundance was contrary to the findings in the Hikurangi Margin which showed a negative relationship with curvature and profile curvature, and suggests that abundance is not always greatest in seafloor depressions. Sun & Fleeger (1994) showed that recolonization processes and abundance patterns of meiofauna depend on the interaction between the hydrodynamic regime associated with seafloor depressions and the life style of meiofauna (e.g., epibenthic or burrowers), and it is possible that similar interactions influence the abundance patterns of meiofauna in this study region, resulting in different patterns between regions. However, the lower level of taxonomic resolution used in this study prevented further analysis to confirm this result.

The second-stage analysis showed that meiofaunal abundance, diversity and community structure at seep habitats were significantly different from the other habitats in the Hikurangi Margin and the differences in community structure were due to variation in the relative abundances of a large number of taxa rather than the presence or absence of unique taxa. Overall abundance was higher at the seep habitat compared with the other habitats, with nauplii and amphipods contributing most to community dissimilarity. High densities at seep sites compared with the adjacent slope habitat have also been observed previously, and have mainly been due to elevated abundances of nematodes and copepods (Pape et al., 2011; Shirayama & Ohta, 1990; Van Gaever et al., 2006; Vanreusel et al., 2010). In the present study, the high abundance of copepods and nauplii at cold seeps was opposite to the pattern observed by Van Gaever et al. (2009), where low abundances of copepods and nauplii were observed, and kinorhynchs, polychaetes, and gastrotrichs were more abundant. Similarly, the high abundance of nematodes, kinorhynchs and ostracods in canyon and slope habitat compared with seep habitat was different to patterns observed elsewhere (Van Gaever et al., 2006). Priapulid larvae were only observed in the seep habitat in the Hikurangi Margin, and the reason for this observed pattern remains unclear. In the present study, diversity was higher in the seep compared with other habitats, which were similar to each other. This finding is similar to Bianchelli et al. (2010), where canyons and slopes were equally diverse, but opposite to other studies where seep diversity was lower than canyon and slope habitats (Ingels et al., 2009; Van Gaever et al., 2009).

Our results support the general findings that there is an effect of seeps on meiofaunal abundance, diversity and community structure (Lampadariou et al., 2013). Higher meiofaunal abundance at seeps has been attributed to high food availability, resulting from methane seepage fuelling bacterial productivity (Van Gaever et al., 2006); a number of nematode and copepod species are adapted to exploiting bacteria in sediment patches with high methane levels (Zeppilli et al., 2011). In addition, a broad range of geological and sedimentary structures (e.g., gas seepage, microbial mat, pockmarks), and seep epifauna generate habitat (e.g., tubeworms, mussels, clams), resulting in habitat heterogeneity, both above and below the sediment surface (Judd, Jukes & Leddra, 2002; Levin, 2005). This habitat heterogeneity is likely to be a key reason for the relatively high diversity in seep habitats in the Hikurangi Margin, where microbial mats, sediment patches contained methane/hydrogen sulphide, clam beds, and carbonate structures have been observed (Baco et al., 2010). Increased microhabitat heterogeneity at seeps compared to other adjacent deep-sea habitats provides a broad array of geophysical environments including those that some fauna are particularly adapted to, such as nematodes that occur in the oxygenated sediment underneath siboglinid tubeworm patches (Vanreusel et al., 2010). Each seep site is unique with different geophysical structure, and thus the influence of the seepages on benthic biodiversity is likely to be site-specific (Zeppilli, Canals & Danovaro, 2012). This proposition could explain the different responses of meiofaunal taxa in our study to those in previous studies (Pape et al., 2011; Van Gaever et al., 2009; Van Gaever et al., 2006).

Meiofaunal community attributes in surface and subsurface sediment layers

The magnitude of sediment depth-related differences in meiofaunal community attributes was substantially greater than for habitats or water depths. This finding is consistent with Ingels & Vanreusel (2013) who observed that variability in meiofaunal communities between sediment depth layers was much greater than variability observed at larger geographical scales (10–100 km).

Meiofaunal abundances were much higher in the surface than the subsurface layer of the sediments, except for nematodes which showed the opposite trend. These results are comparable with findings from other meiofaunal studies, where abundance decreased with sediment depth, and where nematodes become the dominant taxon at subsurface depths (Danovaro, Gambi & Della Croce, 2002; Ingels et al., 2009; Neira et al., 2001). In the Hikurangi Margin region, the differences in the abundance in surface and subsurface sediment layers were greater between canyon and seamount habitats, and between shallow and deep sites. This result may be explained by the complex hydrodynamic regime associated with canyons that can affect the deposition and accumulation rates of sediments and organic matter, resulting in a pronounced structuring of the sediment column within the canyon. Abundance in surface sediment increased with water depth, while abundance in subsurface sediment decreased with water depth, a pattern similar to that observed by Vanaverbeke et al. (1997). These authors argued that the low input of organic matter at the deeper sites, as well as shallow penetration of organic matter in the sediment due to lower bioturbation, could explain this pattern (Vanaverbeke et al., 1997).

Different factors may be driving variation in the abundance of surface and subsurface meiofaunal communities in different regions. In the Hikurangi Margin, seafloor depressions apparently contributed to higher abundance in surface sediment than on flat or elevated ground, by reducing current flow and helping deposition and meiofauna settlement (Fleeger, Yund & Sun, 1995; Giere, 2009). Changes in hydrodynamic conditions around seabed features may also affect larval settlement and sediment grain size characteristics (Butman, 1987; McClain & Barry, 2010). In the subsurface sediments, abundance was positively correlated with sediment phaeopigment concentration and standard deviation of the slope (a proxy measure for slope roughness). Enhanced food availability in the sediment, as indicated by elevated phaeopigment concentrations derived from surface water productivity, has frequently been shown to support higher meiofaunal abundance (Ingels et al., 2009; Pusceddu et al., 2009). It remains unclear how slope roughness is likely to influence meiofaunal abundance in subsurface sediment.

In the Bay of Plenty, abundance was positively influenced by surface chlorophyll concentration and plan curvature in the surface sediments. In the subsurface sediment, abundance was positively related with both sediment organic carbon content and kurtosis. Increased organic carbon content in the sediment has been shown to favour elevated meiofaunal abundance (Ingels et al., 2009; Morse & Beazley, 2008). Sediment kurtosis is a measure of the particle size distribution, and high values of kurtosis indicate that there are outliers in the distribution (heavy-tailed relative to normal distribution), and could therefore be interpreted as a measure of habitat heterogeneity. Similar proxies of sediment heterogeneity have been shown to influence meiofaunal abundance (Netto, Gallucci & Fonseca, 2005), because habitat heterogeneity increases the partitioning of food resources (Levin et al., 2001; Whitlatch, 1981).

Meiofaunal diversity was higher in the surface than subsurface sediment layer in both regions. Similarly, Vanaverbeke et al. (1997) and Danovaro, Gambi & Della Croce (2002) found diversity was typically highest in surface sediment and decreased in deeper sediments, where nematodes become the dominant taxon. In the Hikurangi Margin, diversity in surface sediments was negatively influenced by particle size diversity, mean particle size, and fishing. Negative relationship between these variables and diversity may be an indication of indirect effect of trawling, as noted earlier. The diversity of subsurface meiofauna was not correlated with trawling intensity, but was greater in seafloor depressions than on flat or elevated ground. As mentioned earlier, seafloor depressions may increase meiofauna settlement and deposition of organic matter due to reduced water flow, and increased food availability may enhance diversity (Lambshead et al., 2000). The different patterns observed between surface and subsurface sediment layers in the Hikurangi Margin may reflect the greater exposure of surface communities to the direct and indirect effects of trawling. In the Bay of Plenty surface sediment, surface chlorophyll concentration was positively correlated to diversity, while water depth was negatively correlated to diversity. Surface chlorophyll concentrations provide an indication of the flux of organic matter and phytodetritus to the sea floor, and diversity can increase with an increase in organic flux (Lambshead et al., 2000). A decrease in diversity with increased water depth is possibly related to decreased food availability with depth (Vanaverbeke et al., 1997). In the Bay of Plenty subsurface sediment, kurtosis and sediment organic carbon content were positively correlated to diversity. As described above, these findings are consistent with increased habitat heterogeneity increasing the partitioning of food resources (Levin et al., 2001; Whitlatch, 1981), and increased organic carbon content in the sediment has been shown to increase diversity (Lambshead et al., 2000). Thus, in the Bay of Plenty, meiofaunal diversity in both surface and subsurface sediments were positively linked with proxies of food availability. It remains unclear why a positive correlation between kurtosis and diversity was only found in subsurface sediments.

Meiofaunal community structure was different between surface and subsurface sediment in both Hikurangi Margin and Bay of Plenty. Nauplii, copepods and kinorhynchs were the highest contributors to community dissimilarity between sediment depths for both regions, where the abundance of these taxa was higher in the surface than in the subsurface sediment. Nauplii and copepods are generally the second most abundant taxa after nematodes in the sediment (Danovaro, Gambi & Della Croce, 2002; Vanaverbeke et al., 1997). Typically, copepods and kinorhynchs occupy the well oxygenated sediment layer and are more sensitive to low oxygen concentrations than nematodes (Grego et al., 2014; Vidakovic, 1984), which may be the reason for their higher abundance in the surface than subsurface sediment. The higher abundance of kinorhynchs may also be related to higher food availability in surface sediments, since kinorhynch abundance has a positive relationship with food availability (Shimanaga, Kitazato & Shirayama, 2000). In the Hikurangi Margin region, profile curvature was most highly correlated with meiofaunal community structure in the surface sediment, whilst curvature and phaeopigment were most correlated with community structure in the subsurface sediment. As already noted, seafloor depressions tend to accumulate organic matter and increased meiofaunal settlement from the water flow than elevated slope, and thus influence community structure. The greater importance of phaeopigment for the subsurface community compared to surface community may be due to the generally low food availability in subsurface sediment layers, where limited shifts may have relatively strong effects on communities (Giere, 2009). In the Bay of Plenty, surface chlorophyll concentration was correlated most to community structure in the surface sediment, while kurtosis (a measure of habitat heterogeneity) was correlated most to community structure in the subsurface sediment. These results largely reflect similar patterns observed for abundance and diversity in the region.

Relative vulnerability of meiofauna communities to anthropogenic disturbance

Clear differences in meiofaunal community attributes between the two study regions imply potential regional differences in vulnerability to disturbance caused by bottom trawling, and other physical disturbances that may impact upon the seafloor in the future, such as seabed mining. Bottom trawling have been associated with sediment physical characteristics modification, reducing the availability of food within the sediment and altering habitat characteristics (Martín et al., 2014; Pusceddu et al., 2014). These impacts have been linked with changes in meiofauna abundance and diversity, although not necessarily declines in these community measures (Hinz et al., 2008; Pusceddu et al., 2014). However, short-term microcosm experiment on the effect of disturbance on deep-sea nematode colonisation on enriched and unenriched sediments showed that nematode abundance and diversity were significantly higher in the enriched sediment, suggesting that the presence of food can enhances meiofaunal recolonization, and resilience to disturbance (Gallucci et al., 2008). Thus, meiofauna communities in the Hikurangi Margin, which experience higher surface water productivity and related food availability in the sediment, could be less vulnerable to the effects of disturbance (from bottom trawling or seabed mining) than those in the lower food availability sediments of the Bay of Plenty (Leduc et al., in press).

Within regions, abundance was the only community attribute that differed significantly among habitats. The higher abundance at canyon habitats implies that the vulnerability of canyon communities to anthropogenic disturbance may be different from that of other deep-sea communities. Canyon communities differed from seamount and slope communities due to differences in the abundance of a number of shared taxa, but the former communities also supported slightly more rare taxa than slope and seamount communities. The presence of rare taxa can make a community more susceptible to disturbance when they occur at low densities, as it reduces the chances for successful recolonization, making them potentially vulnerable to localised extinction events. Canyon communities might also be more vulnerable to bottom trawling than other communities because of the generally steep topography of canyon habitats, which makes them prone to slope instability and turbidity flows following trawling events (Puig et al., 2012). This instability can have direct negative impacts on canyon meiofauna, since increased turbidity and sedimentation rates may cause sudden burial of infauna, and slope instability can removed organic-rich sediment down-slope to deeper parts of the canyon (Puig et al., 2012; Pusceddu et al., 2014). Conversely, this organic matter enrichment from the upper canyon might favour meiofauna at deeper locations. Nevertheless, other physical characteristics of canyons, such as the presence of hard substrates and complex topography may protect areas of soft sediment from physical disturbance, providing a source for faunal recolonization to disturbed areas of the canyon (Puig et al., 2012).

The inclusion of seep habitat in the among-habitat comparison in the Hikurangi Margin showed that community abundance, diversity and community structure in seeps were different compared with canyon, slope and seamount habitats. Meiofauna seep communities maybe more vulnerable to disturbance because seep habitats: (1) have complex geological morphology and biogenic structures that increase the microhabitat heterogeneity, which in turns supports a distinct and diverse meiofaunal community that includes temporary meiofauna (such as priapulid larvae that were observed only at the seep habitat in this study); (2) the relatively small and localized seep microhabitats and the sometimes large distances between habitat patches (Greinert et al., 2010), can reduce chances for successful recolonization following anthropogenic disturbance and make seep communities potentially vulnerable to localised extinction events; and (3) potential modification of fluid flow patterns resulting from future large-scale extraction of methane hydrates might affect the persistence or structure of seep communities (Baco et al., 2010; Bowden et al., 2013). Seep habitat and megafauna in the Hikurangi Margin are known to have already been subjected to fishing impacts, and could be subjected to drilling for hydrates in the future (Baco et al., 2010; Bowden et al., 2013).

Clear differences in meiofaunal communities between surface and subsurface sediment layers also imply a relative vulnerability to disturbance, such as from bottom trawling or seabed mining. Bottom trawling can cause widespread damage to sediment column by increased sediment resuspension and deposition, sediment particle size alteration and reduced food availability within the sediment, including changes in oxygen penetration depths (Martín et al., 2014). These habitat modifications can have an impact on infauna (Pusceddu et al., 2014; Schratzberger et al., 2009), and likely to be greater for fauna inhabiting surface of the sediment. Copepods and kinorhynchs, for example, are generally more prevalent in surface than subsurface sediment (Grego et al., 2014; Shimanaga, Kitazato & Shirayama, 2000), making them vulnerable to disturbance that may only affect the sediment surface. In contrast, nematodes can penetrate deeper into the sediment (up to 50 cm depending on sediment types) as they are more tolerant of low oxygen concentrations (Grego et al., 2014; Moens et al., 2014) and may therefore avoid some of the impacts. However, previous studies also found large differences in sediment compaction between untrawled and trawled areas, where surface sediment at trawled areas are much denser, which may affect the nematodes abilities to penetrate deeper in the sediment column (Martín et al., 2014; Pusceddu et al., 2014). Meiofauna may also be resuspended by physical disturbances, instead of being killed directly because of their smaller sizes, and can quickly recolonize the sediment column. Copepods can rapidly recolonise sediments via active dispersal in the water column, while nematodes can only recolonise sediment directly from adjacent undisturbed sediment or through suspended sediment transport (Schratzberger et al., 2004). Nematodes can withstand disturbance and recover faster than other sediment inhabiting meiofaunal groups subjected to disturbance, probably due to their high abundance and short generation time (Schratzberger, Dinmore & Jennings, 2002; Sherman & Coull, 1980), and may thus dominate surface sediment meiofauna communities following trawling (Schratzberger, Rees & Boyd, 2000). Deep-sea mining when it occurs could also cause disruption to the seafloor, but this is likely to result in surface and subsurface sediment meiofauna being equally vulnerable to disturbance. While some deep-sea minerals are found predominantly on the seafloor surface (e.g., phosphate and manganese nodules), present designs for mining tools are expected to disturb at least the upper 5–10 cm layer of soft sediment, and impact the meiofauna to this depth (Miljutin et al., 2011). Similarly, impacts from certain types of trawling will penetrate well into the subsurface layer (Martín et al., 2014). Thus, it is clear that meiofaunal communities are vulnerable to disturbance, and living deeper in the sediment does not necessarily offer protection.

Supplemental Information

Supplemental Information 1 List of final environmental variables that were used in the DistLM analysis and the correlated variables that were removed prior to analysis of meiofauna for Hikurangi Margin and Bay of Plenty region.

[STD = Standard deviation of depth based on 3, 5, 7, 15 grid cell focal mean, Slope STD = Standard deviation of slope, Vrm = terrain rugosity, range = depth range, curvature = change of the slope, profile curvature = curvature of the surface in the direction of the slope, plan curvature = curvature of the surface perpendicular to the slope direction].

Click here for additional data file.

Supplemental Information 2 SIMPER analysis results showing environmental variables accounting for regional dissimilarity between the Hikurangi Margin and Bay of Plenty study regions (cut-off applied at 70% contribution).

[Av.Value = average environmental variable value, Av.Sq.Dist = average dissimilarity, Sq.Dist/SD = Dissimilarity/Standard Deviation, Contrib% = % contribution to overall dissimilarity, Cum.% = % cumulative dissimilarity]. Higher average value are shown in bold.

Click here for additional data file.

Supplemental Information 3 Results of PERMANOVA analysis test for the effects of habitat, water depths, sediment depth and their interaction on meiofaunal abundance at the Hikurangi Margin and Bay of Plenty study region.

Click here for additional data file.

Supplemental Information 4 Results of PERMANOVA analysis test for the effects of habitat, water depths, sediment depth and their interaction on meiofaunal diversity at the Hikurangi Margin and Bay of Plenty study region.

Significant factors at the 5% level are shown in bold. [df = degrees of freedom, SS = sum of squares, MS = mean square, Pseudo-F = Pseudo-F statistic, P = Probability, Unique perms = number of unique permutations, √ECV = square root of estimates of components of variation].

Click here for additional data file.

Supplemental Information 5 Results of PERMANOVA analysis test for the effects of habitat, water depths, sediment depth and their interaction on meiofaunal community structure at the Hikurangi Margin and Bay of Plenty study region.

The effect of spatial covariates were taken into account for community analysis (result not shown). Significant factors at the 5% level are shown in bold. [df = degrees of freedom, SS = sum of squares, MS = mean square, Pseudo-F = Pseudo-F statistic, P = Probability, Unique perms = number of unique permutations, √ECV = square root of estimates of components of variation].

Click here for additional data file.

Supplemental Information 6 Results of second-stage analysis for stations from 700, 1,000 and 1,200 m water depth strata showing meiofaunal abundance and diversity between habitats and sediment depths for Hikurangi Margin.

[Abundance shown in total meiofauna individuals per 10 cm2; diversity as meiofaunal taxon richness].

Click here for additional data file.

Supplemental Information 7 Results of PERMANOVA analysis tests on second stage analysis. Results showed the effects of habitat (slope, canyon, seamount and seep), water depths (700 m, 1,000 m, 1,200 m), sediment depth and their interaction on meiofaunal community structure at the Hik.

Significant factors at the 5% level are shown in bold. [df = degrees of freedom, SS = sum of squares, MS = mean square, Pseudo-F = Pseudo-F statistic, P = Probability, Unique perms = number of unique permutations, √ECV = square root of estimates of components of variation].

Click here for additional data file.

Supplemental Information 8 Raw data for regional analysis.

Data used for PERMANOVA analysis on community attributes between Hikurangi Margin and Bay of Plenty.

Click here for additional data file.

Supplemental Information 9 Raw data for PERMANOVA analysis at each region, Hikurangi Margin and Bay of Plenty.

Data used for PERMANOVA analysis on community attributes at each region Hikurangi Margin and Bay of Plenty. Data for second stage analysis at Hikurangi Margin sites also included.

Click here for additional data file.

Supplemental Information 10 Raw data for DistLM analysis for each region, Hikurangi Margin and Bay of Plenty.

Click here for additional data file.

We thank the scientific personnel of voyages TAN0616, TAN1004, and TAN1206, and the officers and crew of RV Tangaroa. We are also grateful to Lisa Northcote and Karen Thompson (NIWA) for sediment analyses, and Andy McKenzie (NIWA) for help in R software.

Additional Information and Declarations

Competing Interests

Author Contributions

Field Study Permissions

Data Deposition

The authors declare that they have no competing interests. Katrin Berkenbusch is an employee of Dragonfly Data Science, Wellington, New Zealand.

Norliana Rosli performed the experiments, analyzed the data, wrote the paper, prepared figures and/or tables.

Daniel Leduc conceived and designed the experiments, analyzed the data, contributed reagents/materials/analysis tools, wrote the paper, reviewed drafts of the paper.

Ashley A. Rowden conceived and designed the experiments, wrote the paper, reviewed drafts of the paper.

Malcolm R. Clark conceived and designed the experiments, reviewed drafts of the paper.

P. Keith Probert contributed reagents/materials/analysis tools, reviewed drafts of the paper.

Katrin Berkenbusch performed the experiments, analyzed the data, reviewed drafts of the paper.

Carlos Neira reviewed drafts of the paper.

The following information was supplied relating to field study approvals (i.e., approving body and any reference numbers):

The samples were collected under Special Permit (542) issued by the Ministry for Primary Industries pursuant to section 97(1) of the Fisheries Act 1996.

The following information was supplied regarding data availability:

The raw data has been supplied as a Supplemental Dataset Files.

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
