# Peer review of "Differences in meiofauna communities with sediment depth are greater than habitat effects on the New Zealand continental margin: implications for vulnerability to anthropogenic disturbance"

_PeerJ, doi:10.7717/peerj.2154_

## Round 0.1 · original submission · Major Revisions

I have now heard back from three reviewers. All were generally positive about your manuscript, with reviewers 1 and 2 recommending some changes here and there (additional references, and some revision/deletion of figures. Most importantly, reviewer 3 offers some very insightful and helpful ideas on your analyses. These suggestions may result in additional or redoing of analyses, and extensive rewriting of sections. In particular, reviewer 3's comments on being careful to not overstate conclusions bears careful consideration.

Therefore, based on these reviews, my decision is 'major revision'.

Reviewer 1 ·

Basic reporting

This manuscript, written in a fluent and easily readable English, provides an in-deep analysis of meiofaunal community changes across different deep-sea habitats from the continental slope off New Zealand. I have no major issues about the text, apart I would encourage the authors to be consistent here and there with the terminology (e.g., “profile curvature” in the main text is “plain curvature” in Table 6). This manuscript represents one among the very few studies investigating deep-sea meiofaunal communities features at multiple spatial scales, from the micro- (among sediment depths) to the regional scales. The Introduction provides a concrete and synthetic description of the most important and recurrent geomorphological features along the continental margins and well describes the ecological importance of meiofauna in the deep sea. The authors have clearly and satisfactorily anticipated the poorness of previous data related with meiofaunal attributes from different deep-sea habitats, thus naturally bringing the reader to their focus, that is the variability of meiofaunal features at varying spatial scales. In this context, the authors have also provided some clues about the potential effects of deep-sea trawling: this is, in my opinion, of particular interest and timely with respect of management actions currently being under decision in most countries where deep-sea trawling is present or even increasingly carried out (at progressively increasing water depths). In this regard, I have just to notice that some of the references about the impacts of bottom trawling on meiofauna reported in the introduction deal with shallow sites (eg Pranovi et al, Shratzberger et al: lines 87-99). Since this manuscript focus on the deep sea, I would avoid mixing up results from shallow habitats also because sometimes the impacts at shallow sites are not as clear (if not opposite to) as those reported from the deep-sea. The objectives of the study are fully and clear reported in the last sentence of the introduction. However, for clarity reasons, I would see a description of the formal (null) hypotheses fitting the experimental design reported in the methods. The structure of the article conforms with the journal style. Figures 1-5 and Figure 8 are useful. I must notice that using interpolating lines for displaying regressions (not correlations!) results in Figures 6-7 should be carried out only if the results of marginal tests in the DISTLM procedure were carried out using linear regressions. This information is not provided in the methods. However, even in that case, since the regressions (in terms of explained variance) appear overall very weak, I would omit Figures 6-7, also because these simply provide a graphical representation of data that are shown in Table 6. Raw data have been made available as supporting files.

Experimental design

As far as I can see, data of meiofauna are apparently used for the first time in this manuscript. Some of the data reported in this manuscript (dealing with characteristics of the investigated area) have been already reported elsewhere, as are apparently data dealing with macro- and megafauna. For transparency, I would like to see a declaration from the authors stating that the data on meiofauna are reported here for the first time or, eventually, clearly identifying what data have been already published or are under evaluation elsewhere and a clear identification of environmental data sources if effectively already published elsewhere.
The sampling strategy is sufficiently robust to support the work hypotheses, the statistical tools are appropriate and the statistical analyses are clearly described.

Validity of the findings

I think the authors have done a honest and robust analysis of their data, using a well-mixed combination of formal tests and correlation approaches which allowed them to provide a good representation of spatial variations of meiofaunal attributes in a geomorphological complex area. In my opinion, they have been able also to identify some of the most important drives of variations, which include habitat features, environmental variability and bottom trawling.

Additional comments

I would like to congratulate the authors for having been able to condense a huge amount of data in a very clear and compelling manuscript. I hope my minor suggestions will find them positive and that they will be able to cope with them easily.

Reviewer 2 ·

Basic reporting

I felt that the manuscript was written very well and that it meets the basic reporting standards of PeerJ.

My only concern with the manuscript was that it had many parallels with Baguley et al. (2006) (Metazoan meiofauna abundance in relation to environmental variables in the northern Gulf of Mexico deep sea. Deep-Sea Research I 53: 1344-1362), yet Baguley et al. was not cited by the authors. It seems that several aspects of the current manuscript are corroborated by Baguley et al. (2006), so citing this article is highly necessary.

Experimental design

This study had a sound experimental design.

Validity of the findings

I felt that the data were robust and that the conclusions presented by the authors were consistent with the data.

Additional comments

Great study! I look forward to seeing it in print.

·

Basic reporting

No comments

Experimental design

No comments

Validity of the findings

Line 187:
The authors used the surface production derived from NASA SeaWiFS. But, as sinking into the water column, the organic matter was consumed by plankton and microbe and small portion of organic matter produced at ocean surface is supplied to the deep-sea sediments. I suggest they use estimated POC flux to the sediment at each station, proposed by Lutz (2007, Journal of Geophysical Research), instead of surface productivity its self.


Line 210:
It was unclear whether they examined meiofaunal community attributes for each core or for each station. They should clarify.


Line 274:
The authors should present sufficient evidences why these four variables showed ‘substantial’ variability between two regions. I suggest they show the table of the result of SIMPER analysis, like Table 2.


Line 284:
In total, 12 taxa were identified in the Hikurangi Margin, while 14 taxa in the Bay of Plenty. But, on average, 7 taxa per core in the Hikurangi Margin, while 6 taxa per core in the Bay of Plenty. The authors should clarify which region have higher diversity.


Line 418:
The authors discussed the trawling activity could affect the differences in meiofaunal attributes between two regions at this paragraph. But, there are also large differences in surface productivity, sediment phaeopigment concentration and organic carbon content of the sediment between the two regions (discussed at the above paragraph). These variables (related to the productivity in each region) would have more effect on the meiofaunal assemblages. Moreover, they did not examine the statistical test including two regions. Therefore, it’s hardly possible that they could detect the responsible environmental variables to the difference in meiofauna assemblage in two regions. In nature, they could not discuss on the trawling effect on meiofauna assemblage without meiofaunal information before and after the disturbance. The relationship between biological community and anthropogenic disturbance is a one of the hottest topics. Therefore, the authors should carefully discus on this topic.


Line 595
At this paragraph, the authors discussed the difference in vulnerability of meiofaunal community between the two regions. But, it was not clear whether they wanted to discuss on the disturbance in the past (trawling) or that in the future (seabed mining).


Line 606
As the authors noted, the difference in vulnerability to disturbance is closely related to the level of surface productivity. This point would be the basis of discussion on difference in vulnerability between the two regions. I suggest the authors stated the difference in surface productivity between the two regions at the top of this paragraph.

---

## Round 0.2 · accepted · Accept

Both reviewers have offered feedback on this latest version, and find it to be well revised, answering all concerns they had. Congratulations!

Reviewer 1 ·

Basic reporting

This manuscript, slightly modified after a first round of review, provides a well-conceived analysis of meiofaunal community changes across different deep-sea habitats from the continental slope off New Zealand. The authors have been able to cope convincingly with all my minor comments (by the way, I must apologize for the misinterpretation of profile vs. plain curvature interpretation) either in the main text or the rebuttal letter.
As I noticed in my previous review report, this manuscript, that investigates deep-sea meiofaunal communities features at multiple spatial scales, deserves to be published. Very appreciable and timely (because of the world race to ban deep-sea trawling at least below 800 m depth) is the analysis of the potential effects of deep-sea trawling on the observed patterns. Therefore, I have no further objections, and I strongly support the timely publication of this paper in Peer J, as soon as possible.

Experimental design

The authors have addressed any issue of (eventual) data (re)use in a transparent and compelling manner. As I noticed in my previous round of review, the sampling strategy is sufficiently robust to support the work hypotheses. The statistical tools are appropriate and the statistical analyses are clearly described.

Validity of the findings

I'm still convinced that the authors have done a honest and robust analysis of their data, using a well-mixed combination of formal tests and correlation approaches which allowed them to provide a good representation of spatial variations of meiofaunal attributes in a geomorphological complex area. As noticed above, I have appreciated a lot their attempt to explore signs of impact generated by deep-sea trawling.

Additional comments

I would express my congratulations to the authors for a very well-done job.

·

Basic reporting

The authors have revised very carefully; the answers were reasonable and the revised manuscript reached to the level enough to be published. Therefore, I suggest ‘accept’ for this manuscript.

Experimental design

No Comments

Validity of the findings

The authors revised very efficiently, which make their manuscript more valuable. They answered almost all my comments, and I am satisfied with them. The only one point I don’t agree with is shown below.
The authors noted “using the Lutz proxy is highly unlikely to change the result” for the reason they did not use Lutz model. However, I think this response is nonsense. We cannot know whether the model reads different result or not before the analysis. It is true that Lutz model and chlorophyll concentration (at sea surface) are proxy for the same thing. Nevertheless, Lutz model includes the degradation process during sinking in water column; therefore, the model is more appropriate proxy for the organic matter supplied to the seafloor, which would give more strong evidence for the difference between the two regions. The authors already used the mean value of chlorophyll consentration for the 1997-2010 period on their manuscript. I think it is not so difficult to estimate the POC flux to the seafloor using Lutz’s equation.
However, I don’t insist the authors should use Lutz model if they don’t agree my opinion. The result would not change if they use the Lutz model, as they noted. This is an issue of philosophy/attitude toward the science.